# A hybrid model integrating recurrent neural networks and the semi-supervised support vector machine for identification of early student dropout risk

Huong Nguyen Thi Cam[1,2], Aliza Sarlan[1] and Noreen Izza Arshad[1]

[1] Universiti Teknologi Petronas, Department of Computer and Information Sciences, Seri Iskandar, Malaysia
[2] Software Engineering Department, FPT University, Ho Chi Minh, Vietnam



Corresponding author
Huong Nguyen Thi Cam,
huongntc2@fe.edu.vn

## ABSTRACT

**Background:** Student dropout rates are one of the major concerns of educational institutions because they affect the success and efficacy of them. In order to help students continue their learning and achieve a better future, there is a need to identify the risk of student dropout. However, it is challenging to accurately identify the student dropout risk in the preliminary stages considering the complexities associated with it. This research develops an efficient prediction model using machine learning (ML) and deep learning (DL) techniques for identifying student dropouts in both small and big educational datasets.

**Methods:** A hybrid prediction model DeepS3VM is designed by integrating a Semi-supervised support vector machine (S3VM) model with a recurrent neural network (RNN) to capture sequential patterns in student dropout prediction. In addition, a personalized recommendation system (PRS) is developed to recommend personalized learning paths for students who are at risk of dropping out. The potential of the DeepS3VM is evaluated with respect to various evaluation metrics and the results are compared with various existing models such as Random Forest (RF), decision tree (DT), XGBoost, artificial neural network (ANN) and convolutional neural network (CNN).

**Results:** The DeepS3VM model demonstrates outstanding accuracy at 92.54%, surpassing other current models. This confirms the model's effectiveness in precisely identifying the risk of student dropout. The dataset used for this analysis was obtained from the student management system of a private university in Vietnam and generated from an initial 243 records to a total of one hundred thousand records.

## INTRODUCTION

Student dropout identification has been the most critical task for educational institutes. Student dropout not only have adverse effects on individuals' learning capacity but it also poses a question on creditability of educational institutions making their economy at stake (*Talebi, Torabi & Daneshpour, 2024*; *Tang, Xie & Wong, 2015*). Early prediction

of this risk can be beneficial to educational institutions, students and the nation (*Baars & Arnold, 2014*; *Huynh-Cam, Chen & Lu, 2024*; *Ortiz-Lozano et al., 2020*). Increased dropout rates also increase the potential risk to the productivity of the nation since it results in the shortage of skilled professionals which in turn leads to unemployment. In most cases, students who are planning to dropout do not consider the negative impact of their decisions on their future. In some cases, students do not have access to facilities where they can consult experts for counseling. These negative impacts on society have motivated the researchers and educational institutions to analyze this problem seriously and reduce the dropout count (*El Aouifi, El Hajji & Es-Saady, 2024*; *Queiroga et al., 2020*). In this context, development of an early warning system for prediction of student dropout is of immense importance. With the help of early identification system, students who are likely to drop out of school or college could be recognized and assisted in continuing their schooling (*Sandoval-Palis et al., 2020*; *Santana et al., 2015*).

Regarding the severity of this problem, various educators have suggested the implementation of dropout prediction models which will identify the at-risk students in the early stages (*Berens et al., 2019*; *Jiménez-Gutiérrez et al., 2024*; *Patacsil, 2020*). Literature on dropout prediction models suggested that the learning pattern of the students stored in internal database of educational institutions could be utilized to predict the performance of the students and their risk of attrition (*Iam-On & Boongoen, 2017*; *Jagannath & Banerji, 2023*). Prediction of student's performance at initial stages of the course can be highly effective in controlling student dropout rate. Moreover, prediction of at-risk students in the earlier stages of the course could help in providing personalized assistance in pursuing their studies without any major hurdle (*Prenkaj et al., 2020*; *Queiroga et al., 2022*). However, designing and developing a predictive model for accurately identification of student's behavior based on their learning patterns, especially during the initial stages of the course is a very challenging and complicated task (*López-Zambrano, Lara Torralbo & Romero, 2021*). In general, conventional classroom settings and online learning environments follow a standard process with the same guidelines for all students that introduces more hurdles in providing personalized feedback for each individual student to address specific problems concerning their course or learning path (*Isphording & Raabe, 2019*; *Pek et al., 2022*). Personalized feedback helps the academicians to make timely decisions about providing additional support to at-risk students in preventing their dropout.

Tools such as educational data mining (EDM) have significantly enhanced the current educational process, which has helped the educational institutions to transform the teaching process and make it easy for the students to pursue the course of their choice (*Hegde & Prageeth, 2018*; *Tasnim, Paul & Sattar, 2019*). But these techniques are not effective in recognizing the students at risk in the initial stages which restricts their performance in real-time educational platforms. The drawbacks of these techniques can be addressed by deploying automated artificial intelligence (AI) based machine learning (ML) and deep learning (DL) models (*Chung & Lee, 2019*; *Mduma, Kalegele & Machuve, 2019*). Machine learning (ML) and deep learning (DL) models are highly effective at predicting

student dropout risks during the early stages of their academic journey. By analyzing students' learning patterns, these models can accurately assess performance and provide valuable insights into their learning behavior (*Adnan et al., 2021*; *Kukkar et al., 2024*). The emergence of ML and DL techniques enables researchers to discover intricate learning patterns that define strengths and weaknesses of students (*Haixiang et al., 2017*). Moreover, these models are capable enough to process huge volumes of data that is generated daily through online learning educational platforms. Predictive models based on ML and DL can offer detailed insights into students at risk of discontinuing their studies, thereby enabling educators to implement timely interventions aimed at preventing or reducing dropout rates (*Bello et al., 2020*; *Shiao et al., 2023*; *Song et al., 2023*).

Despite advancements in dropout prediction, significant challenges remain in early and personalized intervention for at-risk students. Current EDM and ML models struggle to accurately identify students likely to drop out in the early stages, particularly in real-time learning environments. Limited attention to non-academic factors and a lack of personalized feedback further hinder timely intervention efforts. Additionally, comparative analyses across various ML and DL models are lacking, which could offer valuable insights into the optimal models for specific educational contexts. The aim of this research is addressing these gaps to improve dropout prevention strategies and educational outcomes. To bridge the identified research gaps and demonstrate the effectiveness of ML and DL in predicting dropout rates, this study focuses on addressing the following research questions:

RQ1: Which machine learning models, when integrated with deep learning approaches, are most effective for predicting student outcomes across diverse educational settings with limited datasets?

RQ2: What techniques can improve model performance, especially in handling imbalanced datasets and overfitting?

RQ3: How can dropout predictions be used to recommend personalized learning paths for at-risk students?

This study introduces a hybrid model to facilitate early-stage dropout risk prediction, allowing for a comprehensive analysis of both labeled and unlabeled features to achieve improved predictive performance. The specific contributions of this research are outlined as follows:

i) A hybrid model, namely DeepS3VM, combining semi-support vector machine (S3VM) with recurrent neural network (RNN) to predict dropout risk at an early stage, enabling robust analysis of both labeled and unlabeled features to achieve superior performance.

ii) Comparision the obtained results using DeepS3VM with ANN, CNN, XGBoost, DT, RF regarding performance of dropout student prediction.

iii) Design a personalized recommendation system (PRS) that suggests individualized learning paths for at-risk students based on the prediction model, designed to support their progress and promote retention.

iv) Illustrate the power of technical in data processing by integrating Apache Cassandra and Apache Spark, offering a scalable and high-performance solution for dataset management and processing in predictive models.

The manuscript is organized in following sections: Literature related to prediction of students' dropout in educational institutions using various techniques has been presented in 'Related Works' section. Proposed methodology covering implementation details of proposed DeepS3VM model and design of personalized recommendation system has been presented in the 'Research Methodology' section. Experimental setup and procedure have been illustrated in the 'Implementation' section. Results of the experiment have been discussed in the 'Results and Discussion' section. The last section concludes the research and presents future directions.

## RELATED WORKS

Student dropout risk and its early prediction is becoming the most focused aspects for educational institutions that is receiving significant attention of researchers all over the world (*Haixiang et al., 2017*). Various strategies have been proposed in literature for early identification of student dropout risks. *Lee & Chung (2019)* implemented a ML based hybrid model by combining Random Forest (RF) and boosted decision tree (BDT) approach to improve the efficacy of student dropout warning system. In this research, an ensemble learning technique combined with a synthetic minority oversampling technique (SMOTE) to address class imbalance problem has been presented. The RF and BDT models have been trained using big data instances and results reported improved performance in terms of receiver operating characteristic curve (ROC) and precision. This helps improve the accuracy and reliability of the student dropout warning system, especially in predicting minority class instances, which are typically harder to identify. However, combining these models may increase complexity lead to increased computational cost and model interpretability challenges. *Adnan et al. (2021)* processed OULAD dataset using six different ML models and one DL model with an accuracy of 91% in predicting students that are at the risk of dropout at initial stages of higher education. It was also suggested in this research that although ML exhibits better performance still there is a need for more in-depth analysis to evaluate the performance of students while using online educational platforms. In particular, early intervention needs to be introduced for catering the dropout risk in online learning platforms that will also motivate the students in maintaining their personalized learning path. This action is helpful in a way that institutes can already start to make prevention while there are still no cases of dropping out and student retention rates are still going up. The mechanism makes a good prediction, but it does not consider all the environmental factors that may lead to student dropout. *Lee et al. (2021)* designed a multilayer perceptron (MLP) based predictive model for prediction of student's dropout risk. This research claimed that proposed MLP based model outperforms other ML models including logistic regression (LR), DT, and naive Bayes (NB) in terms of handling independent variables selected *via* variance analysis. Experimental evaluation showed that MLP model used grades along with extracurricular

activities for identifying the risk and reducing the chances of student dropout. Thus, a better application of DL in the utilization of Big Data that will give a better score is by not only looking at the academic data but also involving the participation of the students in non-academic activities as a factor that might affect their result. In the nutshell, it is more complex for MLPs to explain, it means that it is difficult to display the role of a single variable in the forecast. *McManus (2020)* emphasized early identification of students at dropout risk by presenting a comprehensive approach tailored for everyone that accurately shortlisted target students. This approach assisted the instructors to intervein at initial stages for significant reduction of student dropout rate. This early identification enables timely interventions by instructors. It may require extensive data collection on each student, including personal and behavioral factors, which could raise privacy concerns and increase the complexity of implementation. This could also make scaling the model to larger student populations more challenging. A four- step logistic regression framework has been proposed by *Singh & Alhulail (2022)* for early prediction of student dropout using predictive variables related to socioeconomic scenarios. This research identified that academic and aspiration aspects are the two major things that can help in identifying student-teacher attrition. Researchers reported that for early detection and prediction of dropout rates, it is necessary to consider other crucial factors like student support and counseling to achieve an interactive learning environment with an improved retention rate. A wider focus has been seen to lead to earlier, more comprehensive prediction models that would diagnose efficiently those students who are likely to drop out and would need support. On the negative side, the framework's only use of logistic regression may be such that it stops short of adequately unraveling the deep, intricate relationships among the predictors. If so, this may lead to lower predictability when compared with some other learning models, which are more powerful to tackle such miscellaneous matters.

Many researchers insisted that potential predictive models should start from the beginning of the course to avoid dropouts or attritions. Hence, assisting academia to intervene in students' lives at the right time (*Adnan et al., 2021*; *Lee & Chung, 2019*; *Lee et al., 2021*; *Singh & Alhulail, 2022*). The predictive model developed in *Petegem et al. (2022)* predict student performance in coding courses based on submission behavior for programming exercises. ML model has been trained in this work using history data recorded during learning progress of students. Results show improved accuracy in predicting student success, with an accuracy of 80% near the end of the semester. This model provides valuable insights that aid in the identification of struggling students by the teachers and the subsequent intervention at the right time to improve the success rates of the students. However, this model has the drawback of showing a tendency to achieve higher accuracy of 80% more towards the end of the semester, and thus, it becomes of less utility at the beginning of the semester. RNN model has been developed by *He et al. (2020)* for predicting at-risk students in the online learning environment. This RNN model is integrated with gated recurrent units (GRU) and trained using the OULAD datasets. The results reported improved performance when compared to the accuracy of complex long short-term memory (LSTM) model developed for the same purpose. Most of the times, GRUs are certainly faster and also require fewer computations, thus, they are more

efficient than RNNs in making predictions in real-time the students at risk in online learning environments. The RNNs, in general, may still struggle with long-term dependencies in sequential data. This disadvantage might lead to the model's incompleteness in sketching patterns, which will lead to the failure to grasp the main trends related to the student's dropout risk. A hybrid convolutional neural network (CNN) long short-term memory (LSTM) model (CNN-LSTM) model is deployed by *Tang et al. (2022)* for predicting the dropout rate in Massive Open Online Courses (MOOCs). This hybrid CNN-LSTM model is trained on the KDD Cup 2015 dataset and experimental outcomes show that the incorporation of RNN improves the performance of the CNN in terms of achieving a better accuracy (AUC) score. CNN's feature extraction capabilities together with LSTM's sequential data handling are the two main aspects of the model's better accuracy. This means that the dropout rate prediction issue in MOOCs now has the model operating better (AUC) which detects intricate patterns accurately and from both the static and the temporal data. Though the hybrid (or combination) version outperforms the standalone CNN and LSTM, it is also more complex which would result in higher computational costs and possibly longer training time. The additional complexity might be a problem in developing the model for real-time applications or environments with limited computational resources. *Phauk & Okazaki (2020)* predicted student performance based on a hybrid approach of principal component analysis (PCA) in conjunction with RF, DT, NB and support vector machine (SVM). Using principal component analysis (PCA), RF, DT, NB, and SVM as multiple machine learning models, the dimension of the data is significantly decreased, and at the same time, the predictive power is still retained. This, in other words, is the key to efficiency and accuracy of the model as it keeps on the most important and pertinent features, thus it could be said that the performance of the model is predicted more correctly. One notable drawback is that the use of PCA may result in the deletion of some important information from the original dataset such as say the model. That kind of information should be retained for the correct performance of the model. In addition, the integration of multiple models augments complexity to the system which highly likely may encompass more computational resources and be a lot trickier when it comes to reading the results. Table 1 summarizes the application of ML and DL techniques for predicting students that are at risk of dropout.

Literary review has synthesized that the development of predictive models and early warning systems for controlling student dropout rate in educational platforms could help reduce this problem at earlier stages of their education (*McManus, 2020*). But there are various issues and complexities associated with this process since these models are deployed only for a specific educational platform (*Singh & Alhulail, 2022*). Another prominent issue that hinders the effectiveness of the predictive models is that they are not adaptable to different learning environments, and it is challenging to accompany the diversified requirements of the students (*Petegem et al., 2022*). In addition, the limitation of collected data is also a thorny issue in which predictive models heavily rely on large, accurate, and relevant datasets (*He et al., 2020*). Moreover, most of studies have not yet

Table 1 **Literature review.** The summary of previous researches in terms of methods, dataset, performance, drawbacks, attributes used for prediction.

| Article | Models/Algorithms | Performance | Drawbacks | Attributes used for prediction | Dataset used |
|---|---|---|---|---|---|
| *Adnan et al. (2021)* | RF, SVM, K-NN, ET, AdaBoost, Gradient boosting, ANN and Deep Feed Forward Neural Network (DFFNN) | RF outperformed other models in predicting student performance with accuracy 91% overall. | When applying RF to another educational datasets, student behavior fluctuations can be misinterpreted as significant, leading to overfitting. | Students' demographics, students' Virtual Learning Environment (VLE) interaction, assessments, course registration, and courses offered. | Open University Learning Analytics Dataset (OULAD) |
| *Lee & Chung (2019)* | RF, BDT, and RF with SMOTE | Utilizing SMOTE and ensemble methods to enhance the system's performance, based on the ROC curves and precision-recall. | If the majority class (not-at-risk students) is complex or contains noise, improving the minority class (at-risk students) alone can't be sufficient to achieve high overall model performance because of SMOTE specifically enhances the minority class. | Student's data, human resources, budget, and accounting functions | Samples of 165,715 high school students from the 2014 National Education Information System (NEIS) |
| *Lee et al. (2021)* | LR, DT, NB, and MLP neural network. | The MLP model outperformed remain ones, with the F-score and AUC were 0.87 and 0.98, respectively. | The research lack of focusing the overfitting, computational cost of MLP model. With a large number of parameters and limited training data, MLP can learn noise and irrelevant patterns. | GPA, Age, Gender, Nationality, course registered | Samples of 18,000 students from Brazilian Public University |
| *Singh & Alhulail (2022)* | Logistic Regression Analysis based on Students' attrition | Developing a model to predict and identify at-risk students who may be prone to dropout early in their academic journey. | Logistic Regression may struggle to model interactions between features, which are often crucial in dropout risk analysis. This research did not explicit feature engineering. | Age, gender, marital status, academic grades, parent academic level, social circle of student | Sample of 1,723 student-teachers in public teachers training colleges (TTCs) of a least-developed country (LDC) |
| *Petegem et al. (2022)* | LR, SVM, RF, Stochastic Gradient Descent | A new framework for pass/fail predictions in coding courses based on submission behaviour is presented with accuracy LR model of 80%. | The models were trained on a small or homogenous dataset from one programming course but lack of concern on overfitting. | Age, gender, IQ level, GPA, semester | Sample of 21,000 programming assessments from a private college of Hungary |
| *He et al. (2020)* | RNN, GRU, LSTM | Simple RNN and GRU models outperformed LSTM in predicting student outcomes, achieved over 80% accuracy in identifying at-risk students | The research not fully address how the model handles missing or noisy data, which is common in real-world educational datasets. | Demographics information, course information, mutual information, and assessment performance | Open University Learning Analytics Dataset (OULAD) |

(Continued)

| Table 1 (continued) | | | | | |
|---|---|---|---|---|---|
| Article | Models/Algorithms | Performance | Drawbacks | Attributes used for prediction | Dataset used |
| *Tang et al. (2022)* | CNN-LSTM Network Model | CNN-LSTM model for MOOC dropout prediction, reached the highest accuracy of 94.3% | LSTMs are designed to handle sequential data, but their performance in capturing long-term dependencies can degrade, particularly in very long sequences, which are common in MOOCs that span weeks or months. This could reduce the effectiveness of model. | Students' learning activity logs | KDD Cup 2015 competition |
| *Phauk & Okazaki (2020)* | A hybrid approach combining four baseline ML algorithms with PCA and 10-fold cross-validation is proposed. | SVMRBF, NB, C5.0, and RF are effective algorithms for classification. | In educational settings, student performance metrics such as attendance, engagement, and grades are often correlated. However, this research might assumption all features are independent given the target class, which is rarely true in real-world data. | Score of mathematics of students in the semester I | GDS1 (2,000 samples) and GDS2 (4,000 samples), ADS3 that consists of 1,204 samples. |

recommended learning path for learners to avoid dropout (*Tang et al., 2022*). These issues create a major research gap and there is a need to develop an effective model which can overcome all these drawbacks and provide a robust, adaptable, and flexible model for various courses, educational platforms, and institutions (*Phauk & Okazaki, 2020*). A hybrid ML and DL based model has been proposed in our research that not only mitigates the risk of student's dropout but also provides personalized recommendation system (PRS) to assist students in overcoming their deficiencies and rescue them from getting dropped out from their educational track.

## RESEARCH METHODOLOGY

This study seeks to improve the precision and effectiveness of student dropout risk prediction through the development of a hybrid model focused on identifying and reducing dropout risk. It benefits from predictive nature of ML algorithms and feature extraction of DL algorithms for identifying student dropouts in both small as well as large educational datasets. In addition, the study also intends to recommend personalized learning paths for students at risk of dropping out. A hybrid DeepS3VM model is designed for this purpose which integrates a ML and DL model, namely recurrent neural network (RNN) with semi-supervised support vector machine (S3VM). The proposed DeepS3VM model is capable enough to analyze complex problems using input data samples to identify learning patterns of students. The model is designed to adapt itself to new environments and predicts novel or unknown instances based on previous historical data.

The proposed methodology is presented in the Fig. 1, composed of two distinct subfigures: Figs. 1A and 1B. Figure 1A provides a detailed breakdown of the processes involved in both the data collection and preprocessing stages, offering a clear understanding of the foundation upon which the analysis is based. Figure 1B focuses on the construction of the DeepS3VM model and the personalized recommendation system, demonstrating the key components that drive the model's decision-making. Modules 1, 2, and 3 are included to further elaborate on the methodology, corresponding to the three main sections of Fig. 1. These modules are crucial for a comprehensive understanding of the approach and provide greater insight into the specific methods applied at each stage, thereby ensuring reproducibility and clarity of the proposed framework.

## Module 1: data collection and data preprocessing

This stage outlines the sequential procedures followed to ensure the data is systematically gathered and prepared for further analysis. Most of the educational institutions have their own database wherein the students have registered for different courses. The information from the database provides an insight into the students who are likely to drop out of courses and the university. The data were collected from students who are currently in their second year at a private university in Vietnam. This data includes information related to their academic progress and evaluations of all courses taken during these 2 years of their program. Data privacy standards were strictly adhered to throughout the research. A snapshot of the raw dataset is provided in Table 2, while the dataset specifications are detailed in Table 3.

When using models that are sensitive to incomplete data, removal of missing records can enhance performance stability (*Zhang, Li & Zhou, 2020*). Therefore, after collection of datasets, it has been passed through multiple stages of preprocessing in which duplicate and missing records are removed. Data is augmented and normalized; and significant features are extracted for trend identification that is to be used in prediction of student dropout rate.

## Module 2: building the hybrid DeepS3VM model

The second module of the proposed model deals with training of S3VM and RNN model. Student performance dataset has been applied to train RNN and S3VM models. Then, result of RNN and S3VM model is passed through an averaging function and prediction function for student dropout prediction. The structure of the proposed hybrid model is presented in Fig. 2.

### RNN model

An RNN is a type of artificial neural network designed to identify patterns in sequential data. It can be trained on student performance metrics to predict outcomes like the likelihood of student dropout. As part of the deep neural network family, RNNs excel at learning long-term dependencies, making them ideal for tasks involving sequence prediction. Their strong memory capabilities allow them to capture patterns within sequences, which is why the RNN architecture is utilized in this research to predict the next element in a sequence, such as student dropout rates.

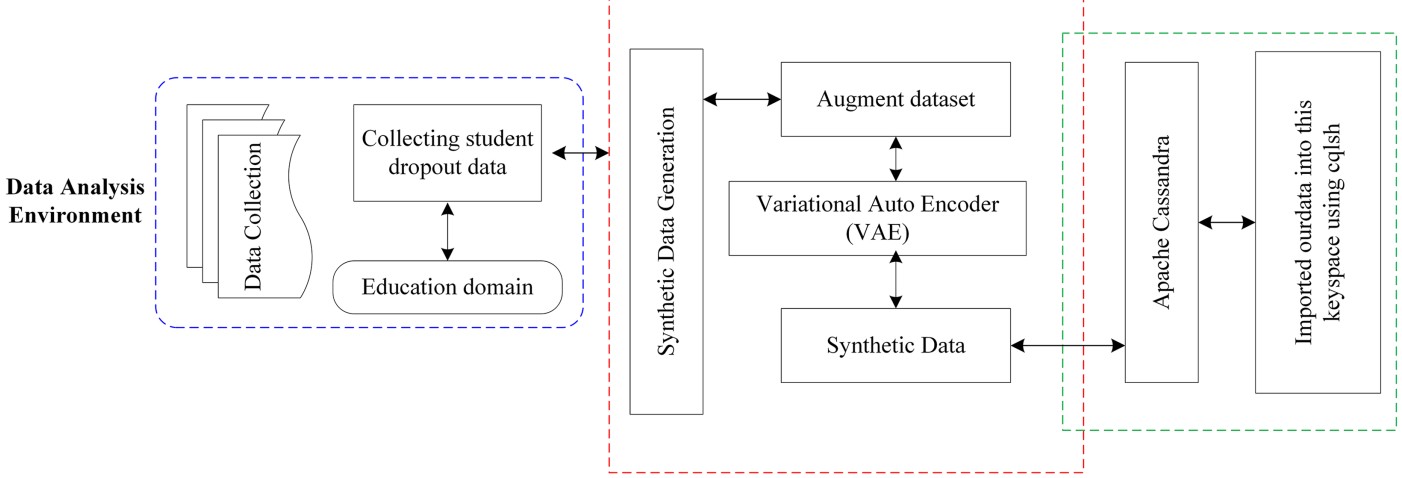

(a) Data collection and preprocessing layer

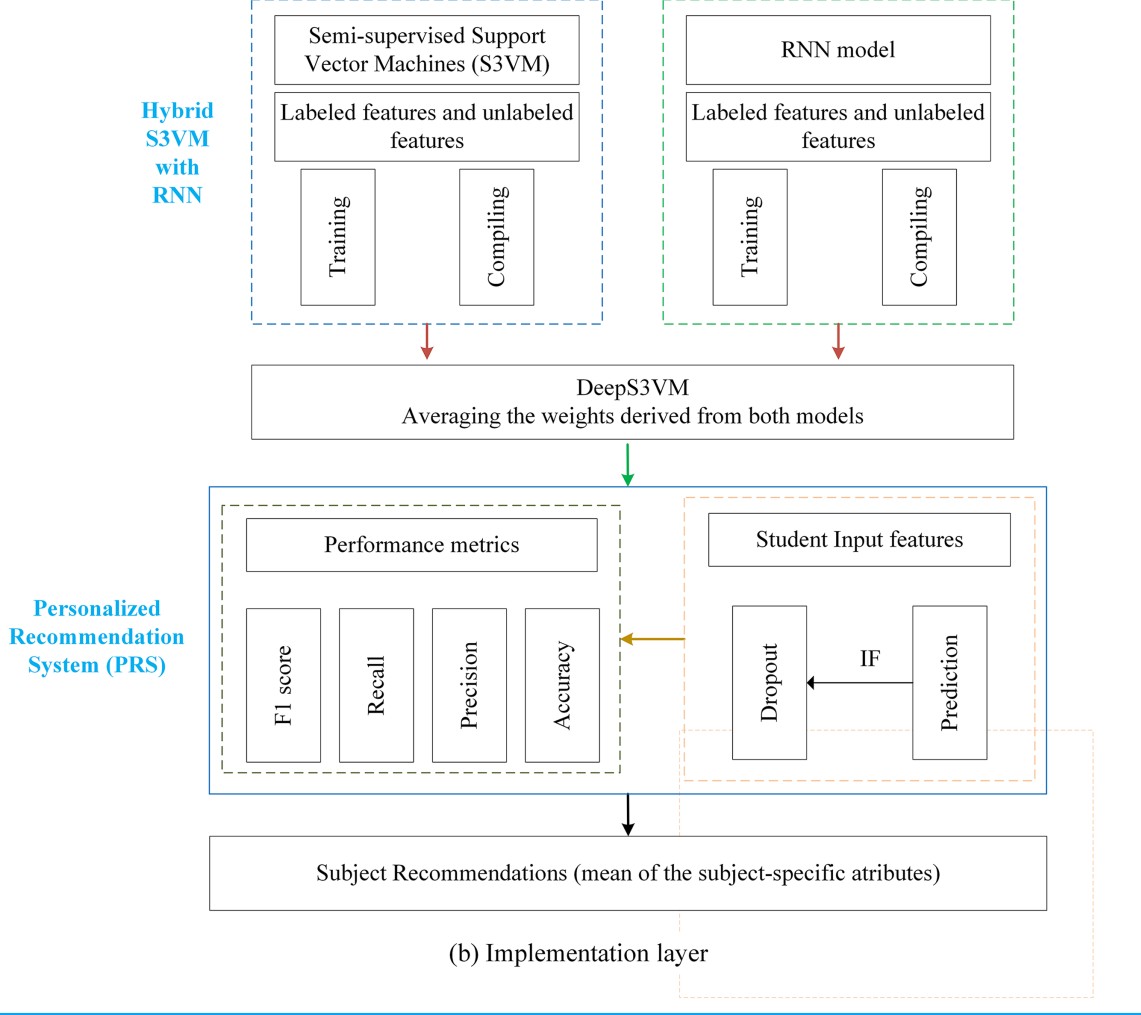

(b) Implementation layer

**Figure 1** **Proposed methodology model.**

**Table 2  Raw data snippet.**

| Code_module | Code_presentation | ID_assessment | Assessment_type | Days attended | Scores |
|---|---|---|---|---|---|
| AAA | 2013J | 1752 | Quiz | 19 | 10 |
| AAA | 2013J | 1753 | Quiz | 54 | 20 |
| AAA | 2013J | 1754 | Assign | 117 | 20 |
| AAA | 2013J | 1755 | Assign | 166 | 20 |
| AAA | 2013J | 1756 | Quiz | 215 | 30 |
| AAA | 2013J | 1757 | Exam | | 100 |
| AAA | 2014J | 1758 | Capstone project | 19 | 10 |
| AAA | 2014J | 1759 | Capstone project | 54 | 20 |
| AAA | 2014J | 1760 | Capstone project | 117 | 20 |
| AAA | 2014J | 1761 | Quiz | 166 | 20 |
| AAA | 2014J | 1762 | Assign | 215 | 30 |
| AAA | 2014J | 1763 | Exam | | 100 |
| BBB | 2013B | 149 | Quiz | 54 | 1 |
| BBB | 2013B | 152 | Assign | 89 | 1 |
| BBB | 2013B | 161 | Assign | 124 | 1 |
| BBB | 2013B | 190 | Capstone project | 159 | 1 |
| BBB | 2013B | 178 | Capstone project | 187 | 1 |

**Table 3  Dataset specification.**

| Aspect | Specification | Description |
|---|---|---|
| Duration | 2020–2022 | – |
| No. of observations | 1,500 | – |
| Features extracted | 1. Student demographics | - |
| | 2. Attendance report | - |
| | 3. Grades | 3. Grade is calculated by aggregating all assessment scores. |
| | 4. Assessment scores | 4. Marks obtained in each assessment present assessment scores. |
| | 5. Status of learning progress | 5. It is measured as " average", " below average" and " above average" from accuracy of the assessment and time taken by the student in completing the task. |
| | 6. Participation in extracurricular activities | 6. It is measured as "Yes" and "No" |
| Dataset size | 3GB | – |

The RNN is particularly suited for this research because it effectively handles sequential data, making it well-suited for predicting academic performance, where student progress over time is viewed as a sequential process. The RNN is trained using parameters such as historical academic records, including grades, attendance, and extracurricular activity participation. Personal student information is excluded from the prediction model as it is deemed irrelevant to the final performance prediction.

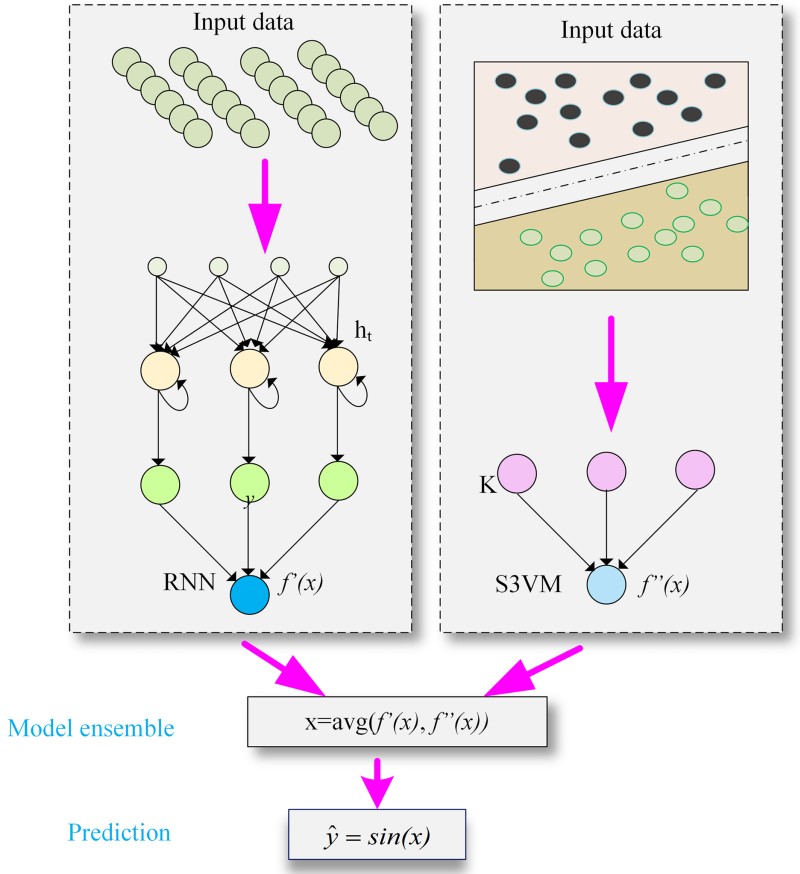

**Figure 2 The structure of DeepS3VM model by combining the characteristics of RNN and S3VM.**

*Lu et al. (2024)* has incorporates advanced preprocessing techniques, including Z-score normalization described in Eq. (1) and feature selection by working on RNN in gen classification. They apply Z-score normalization as part of the data preprocessing step to improve the classification accuracy by ensuring that all input features are standardized to have a mean of zero and a variance of one. This normalization helps to stabilize the training process, particularly for complex datasets such as gene expression profiles, which often have high dimensionality and varying ranges. In our research, the columns containing marks of assessment are normalized into the Z-scale ranging from 0 to 1.

$$z = \frac{x - min(X)}{max(X) - min(X)} \tag{1}$$

$x$: The individual data point that is being normalized
$min(X)$: The minimum value in the dataset or feature $X$
$max(X)$: The maximum value in the dataset or feature $X$.
All remaining missing data are imputed as zero. The large volume of missing data may adversely impact the predictive performance. While various imputation techniques exist,

applying imputation in this case could introduce bias. Furthermore, missing values in academic assessments are likely due to students either not attending exams or failing to submit assignments, making the imputation of zero a reasonable approach. Average final marks have been calculated using a total of hundred marks for each subject. These subject wise average marks obtained are utilized while recommending personalized learning plans to the students who are at the risk of dropout.

A detailed overview of deep learning and its applications in relation to RNNs is provided through Eqs. (2)–(4), which illustrate layers, hidden units, and activation functions, as referenced in the work of *Goodfellow, Bengio & Courville (2016)*. The following sections present a more comprehensive explanation of these functions.

**Layers**: The RNN developed in this study includes three layers of recurrent units to capture temporal dependencies.

$$h_t = \varnothing(W_{hh}h_{t-1} + W_{xh}x_t + b_h) \tag{2}$$

where:

$\varnothing$ is the activation function (ReLU).

$W_{hh}$ is the weight matrix for the hidden state.

$W_{xh}$ is the weight matrix for the input.

$x_t$ is the input at time step $t$.

$h_{t-1}$ is the hidden state from the previous time step.

$b_h$ is the bias term for the hidden state.

Each neuron in one layer is independent from others and connection between neurons has been achieved by stacking two or more layers of RNNs.

**Hidden units**: The number of hidden units in each layer is chosen based on experimentation and validation performance. For the $n^{th}$ neuron, the hidden state $h_{n,t}$ can be obtained by Eq. (3).

$$h_{n,t} = \sigma(w_n X_t + u_n h_{n,t-1} + b_n). \tag{3}$$

The bias associated with neuron $n$ is denoted by $b_n$, while $w_n$ and $u_n$ represent the $n^{th}$ row of the input weight and recurrent weight matrices, respectively. Each neuron is exclusively influenced by the input and its hidden state from preceding time step. Consequently, each neuron in an RNN independently manages a specific spatial-temporal pattern.

**Activation functions**: ReLU activation functions have been used for hidden layers and sigmoid for the output layer, to predict student dropout risk. The output $f'(x)$ at time stamp has been calculated by Eq. (4).

$$f'(x) = \sigma(W_{hy}h_t + b_y) \tag{4}$$

where $W_{hy}$ presents the weight matrix of hidden state to the output, $b_y$ presents bias term for output. The RNN can be viewed as a multi-layer perception applied over time, with shared parameters across time steps. Unlike ordinary RNNs, the proposed model offers a novel interpretation, treating recurrent neural networks as independently aggregating

spatial patterns (represented by $W$) across time (represented by $u$). By stacking two or more layers, the model can exploit the correlations between neurons, as each neuron in the subsequent layer processes the outputs of all neurons from the preceding layer. Output of RNN model is passed to S3VM model for student dropout prediction.

### S3VM model

SVM is a powerful supervised learning algorithm traditionally used for classification and regression tasks. The semi-supervised support vector machine (S3VM) extends SVM to handle semi-supervised learning scenarios, where the model is trained using a combination of labeled and unlabeled data. This can be particularly useful in predicting student dropout, where obtaining labelled data (*i.e.*, knowing which students dropped out) may be difficult, but a large amount of unlabeled data (*i.e.*, students for whom dropout status is unknown) is available.

To develop an S3VM model architecture for student dropout prediction, we need to blend the principles of SVM with semi-supervised learning techniques. The S3VM model has been trained on labeled and unlabeled training data. The model learns to find the optimal hyperplane that separates the two classes (dropout *vs.* non-dropout) by maximizing the margin between them.

The standard SVM objective function focuses on minimizing classification error while maximizing the margin between classes. It seeks to balance two competing goals: minimizing the misclassification of data points and maximizing the distance between the hyperplane and the nearest data points from each class, known as support vectors. Based on the foundation of *Cortes & Vapnik (1995)*, the standard SVM objective functions for binary classification are described in detail and formulated in Eqs. (5) and (6).

$$min_{w,b,\xi} \frac{1}{2}w^2 \; + \; C\sum_{i=1}^{l} \xi_i \tag{5}$$

Subject to

$$y_i(w \cdot x_i + b) \geq 1 - \xi_i, \; \xi_i \geq 0 \tag{6}$$

where:
  $w$ is the weight vector.
  $b$ is the bias term.
  $\xi_i$ are the slack variables to allow for misclassification.
  $C$ is the regularization parameter.
  $y_i$ are the labels (dropout or not dropout).
  $x_i$ are the input features (student data).
S3VM has been originally designed for binary classification. However, S3VMs are being extended to handle multi-class problems through techniques like 'one against all' (OAA) and 'one against one' (OAO). The S3VM objective function defined in our proposed methodology incorporates both labeled and unlabeled data. In this research, $'l'$ denotes the

number of labeled samples and 'u' denote the number of unlabeled samples. The objective function (K) has been calculated in the research of *Joachims (1999)* and described in Eq. (7). The research demonstrated how kernel functions (K) could be used to capture non-linear relationships in the data, allowing SVM to perform well even in complex classification tasks.

$$K = min_{w,b,\xi,\tilde{y}} \frac{1}{2} w^2 + C \sum_{i=1}^{l} \xi_i + C' \sum_{j=1}^{u} \xi_j' \tag{7}$$

For labeled data:

$$y_i(w \cdot x_i + b) \geq 1 - \xi_i, \quad \xi_i \geq 0, \quad \forall i = 1, \dots, l \tag{8}$$

For unlabeled data:

$$y_j(w \cdot x_j + b) \geq 1 - \xi_j', \quad \xi_j' \geq 0, \quad \forall j = 1, \dots, u \tag{9}$$

where:

$\tilde{y}_j$ are the predicted labels for the unlabeled data, $\xi_j'$ are the slack variables for the unlabeled data and $C'$ is the regularization parameter for the unlabeled data in this research. In proposed S3VM, the predicted labels $\tilde{y}_j$ for the unlabeled data are iteratively updated. Initially, these labels can be guessed or set based on some heuristic (*e.g.*, majority class). During optimization, these labels are updated to minimize the objective function.

Now loss function for labeled data and unlabeled data in this research has been calculated in Eqs. (10) and (11) (*Cortes & Vapnik, 1995*). The hinge loss $L_l$ encourages the model to not only classify the data points correctly (with $y_i(w \cdot \phi(x_i) + b) > 0$) but also to maintain a margin of 1 around the decision boundary.

$$L_l = \sum_{i=1}^{l} max(0, \ 1 - y_i(w \cdot \phi(x_i) + b)) \tag{10}$$

$$L_u = \sum_{j=1}^{u} max(0, \ 1 - \tilde{y}_j(w \cdot \phi(x_j) + b)). \tag{11}$$

The study of *Vapnik (1998)* expanded upon the theoretical foundations of statistical learning theory, particularly focusing on the regularization term (R) in the context of Support Vector Machines (SVMs) and general machine learning models. The regularization term plays a crucial role in controlling the complexity of the model and preventing overfitting. The regularization term ensures the complexity of the proposed dropout risk prediction model is controlled using Eq. (12).

$$R = \frac{1}{2} w^2. \tag{12}$$

Final objective function has been developed in this research by combining all above stated functions in Eq. (13)

$$min_{w,b,\xi,\tilde{y}} \frac{1}{2} w^2 + C \sum_{i=1}^{l} max\left(0, \; 1 - y_i(w \cdot \phi(x_i) + b)\right)$$
$$+ C' \sum_{j=1}^{u} max\left(0, \; 1 - \tilde{y}_j(w \cdot \phi(x_j) + b)\right). \tag{13}$$

The decision function in Eq. (14) employed in this study is derived from the foundational work of *Cortes & Vapnik (1995)*. This function computes a value that is used to classify the input $x$. If the result is greater than 0, the point is classified as belonging to one class (*e.g.*, $+1$), and if less than 0, it's classified as the other class (*e.g.*, $-1$).

$$f''(x) = w \cdot \phi(x) + b. \tag{14}$$

$w$: The weight vector learned during training.

$\phi(x)$: The feature vector of the input $x$. In non-linear SVMs, this typically represents the mapping of the input data into a higher-dimensional space *via* a kernel function.

$b$: The bias or intercept term.

## Model ensemble

Finally, to classify whether a student will drop out, this research calculates the average of the final values obtained from both models as represented by $f'(x)$ in Eq. (4) and $f''(x)$ in Eq. (14), the sign of new function will be taken as presented in Eqs. (15a) and (15b).

$$x = avg(f'(x), f''(x)) \tag{15a}$$
$$\hat{y} = sin(x) \tag{15b}$$

where:

$\hat{y} = +1$ (*or* 1) indicates that the student is predicted to drop out.

$\hat{y} = -1$ (*or* 0) indicates that the student is predicted not to drop out.

## Module 3: design of a personalized recommendation system (PRS)

The third module focuses on recommending personalized learning plans for students at risk of dropout, utilizing a personalized recommendation system (PRS). The PRS generates recommendations by averaging subject-specific attributes of at-risk students. In this study, synthetic data is created by augmenting the dataset, which is efficiently processed using big data tools. The hybrid model employed integrates RNN and S3VM, trained on both labeled and unlabeled features, with the PRS leveraging the model's output to generate relevant recommendations. The PRS combines the functional attributes of S3VM and RNN to assess the likelihood of student dropout. Upon predicting dropout risk, the PRS recommends a personalized learning path. When the model identifies at-risk students, their subject-specific averages are compared to the university's predefined threshold for dropout decisions. For students falling below this threshold, the PRS sends a cautionary message along with a customized learning plan, offering assistance in subjects where performance is below the threshold.

The design process of the PRS is outlined in the following steps:

**1. Data preparation:** In the initial stage, the code assembles a new dataset containing an array of essential attributes pertinent to a fresh student. These attributes typically encompass academic performance metrics, such as grades, and other pertinent information pertaining to a variety of subjects.

**2. Feature scaling:** The subsequent process involves feature scaling through standardization, ensuring that all attributes exhibit a common mean of 0 and a standard deviation of 1. This standardization technique is employed to optimize the performance of the DeepS3VM model used in the predictive analysis.

**3. Generating predictions:** Within this phase, the predictive power of the DeepS3VM model is harnessed to evaluate whether the student is prone to dropout. Leveraging ML algorithms, the model assesses the risk associated with a student's academic progression.

**4. Output generation based on prediction**: Following the model's evaluation, the code takes appropriate action based on the predictive outcome. If the model's prediction exceeds or equals a threshold of 0.5, it generates a warning message indicating that "This student is at risk of dropping out." Subsequently, the code proceeds to offer subject recommendations. If the prediction falls below the 0.5 threshold, the code conveys that "This student is not at risk of dropping out," implying that the student's academic performance is relatively secure.

**5. Subject recommendations:** In the event that the student is deemed to be at risk of dropping out, depending on the input data, the system will recommend the subjects that students should learn. Subsequently, it arranges these subjects in descending order based on their mean values.

## IMPLEMENTATION

The pseudocode for implementing DeepS3VM for prediction has been presented in this section.

**Initialization**

**Step 1: Dataset Preparation**

   1.1   *Load the dataset (D).*

   1.2   *Remove labels from the dataset, creating unlabeled data (D_unlabeled).*

   1.3   *Divide the dataset into two sets:*

   - ***Labeled features (D_labeled) for training***:

        *D_labeled = $\{(x_i, y_i)\}$, where i = 1 to N, $x_i$ are feature vectors, and $y_i$ are considered as labels.*

   - ***Further unlabeled features (D_unlabeled) are used for semi-supervised learning:***

        *D_unlabeled = $\{X_j\}$, where j = 1 to M, $X_j$ are feature vectors.*

**Step 2: Construct DeepS3VM Model**

   2.1   *Initialize an S3VM model and a RNN model (L).*

   2.2   *Combine the S3VM and RNN models into the DeepS3VM model:*

$DeepS3VM = \lambda * S3VM + (1 - \lambda) * L$, where $\lambda$ is a hyperparameter controlling the balance between the models.

2.3    *Train the DeepS3VM model using both labeled and unlabeled features:*

**DeepS3VM.train(D_labeled, D_unlabeled)**

**Step 3: Train RNN Model**

3.1    *Initialize a separate RNN model(RNN).*

3.2    *Train the RNN model using only labeled features:*

**RNN.train(D_labeled)**

**Step 4: Prediction Phase**

4.1    *Use the DeepS3VM model to make predictions on the testing data. (D_test):*

**predictions_DeepS3VM= DeepS3VM.predict(D_test)**

4.2    *Use the RNN model to make predictions exclusively on the testing data (D_test):*

**predictions_RNN = RNN.predict(D_test)**

4.3    *Combine predictions by weighted averaging:*

**weighted_predictions = α * predictions_DeepS3VM + (1 − α) * predictions_RNN**, *where α is a hyperparameter controlling*

*the weight of the DeepS3VM predictions.*

4.4    *Generate the final predictions as a balanced assessment of the data:*

**final_predictions = weighted_predictions**

**End**

# Experimental setup

## Software requirements

The experimentation is performed using the latest version of Python (3.11) with Anaconda which is a popular open-source analytics platform. Anaconda distribution helps in creating and launching tools such as Jupyter Notebook. for implementation of combined RNN and S3VM model, the SciKit-learn library and Natural Language Process Tool Kit (NLTK) is used. Machine learning algorithms are deployed in Python using the SciKit module. Jupyter notebook is used for performing experiments. Processor Intel Core i7 with 16 GB RAM and 150 GB hard disk and Windows operating system (64 bit) has been utilized. The proposed DeepS3VM model has been experimentally validated in this section. The experiment has been performed to determine the performance of the proposed adaptive DeepS3VM model. In addition, this research employs a hybrid big data processing approach which is carried out using two effective big data tools namely Apache Cassandra and Apache Spark.

## Hardware requirements

- Operating system (OS): The OS used in this research is compatible with multiple operating systems including Windows (64 bit), Mac and Linux.
- Processor: Intel Core i7 or AMD Ryzen 7

**Table 4 Feature selection for experimentation.**

| Input (Features) | | | | | | Output (Labels) |
|---|---|---|---|---|---|---|
| Demographic information | Socio-economic factors | Psychological and behavioral factors | Academic history | Access to resources | Engagement with learning materials | Academic performance |
| Age | Family income level | Level of engagement in extracurricular activities | Previous educational background (*e.g.*, high school GPA) | Availability of academic support services (*e.g.*, tutoring, counseling) | Frequency of library usage | GPA (Grade Point Average) |
| Gender | Parental education level | Self-reported motivation or commitment to education | Transfer student status (binary: yes/no) | Access to technology (*e.g.*, personal computer, internet connection) | Utilization of online learning platforms | Number of failed courses |
| Ethnicity | Employment status (part-time/full-time) | Perceived social support | Number of semesters completed | Participation in mentorship programs | Participation in study groups or collaborative learning activities | Average grade in major courses |
| First-generation college student (binary: yes/no) | Housing situation (living on-campus/off-campus) | Stress level or mental health indicators (*e.g.*, anxiety, depression) | Number of course withdrawals | | | Number of disciplinary actions |

- RAM: Minimum 16 GB RAM
- Hard Disk: Minimum 150 GB ROM

## Data collection

The student performance dataset was collected from a private university in Vietnam. Most of the data was sourced from the institution's Student Management System (SMS) and Learning Management System (LMS). During the data collection process, strict adherence to the university's data privacy policies was ensured to protect student confidentiality and comply with ethical guidelines.

## Data preparation

Data preparation procedure is divided into multiple stages. The first stage is data cleaning in which missing values and duplicate records are removed. In the second stage, feature extraction has been carried out using an exploratory data analysis (EDA) to uncover patterns, anomalies, relationships, and insight uncovers insights and patterns within the data. After EDA, 26 features (listed in Table 4) have been identified which are correlated to each other.

This process helps in reducing the size of the feature set by including only relevant features. Correlation matrix as shown in Fig. 3 presents '1' being highly correlated and '0' being no correlation at all. Removal of highly correlated features ensures that the data is free from unnecessary redundancy.

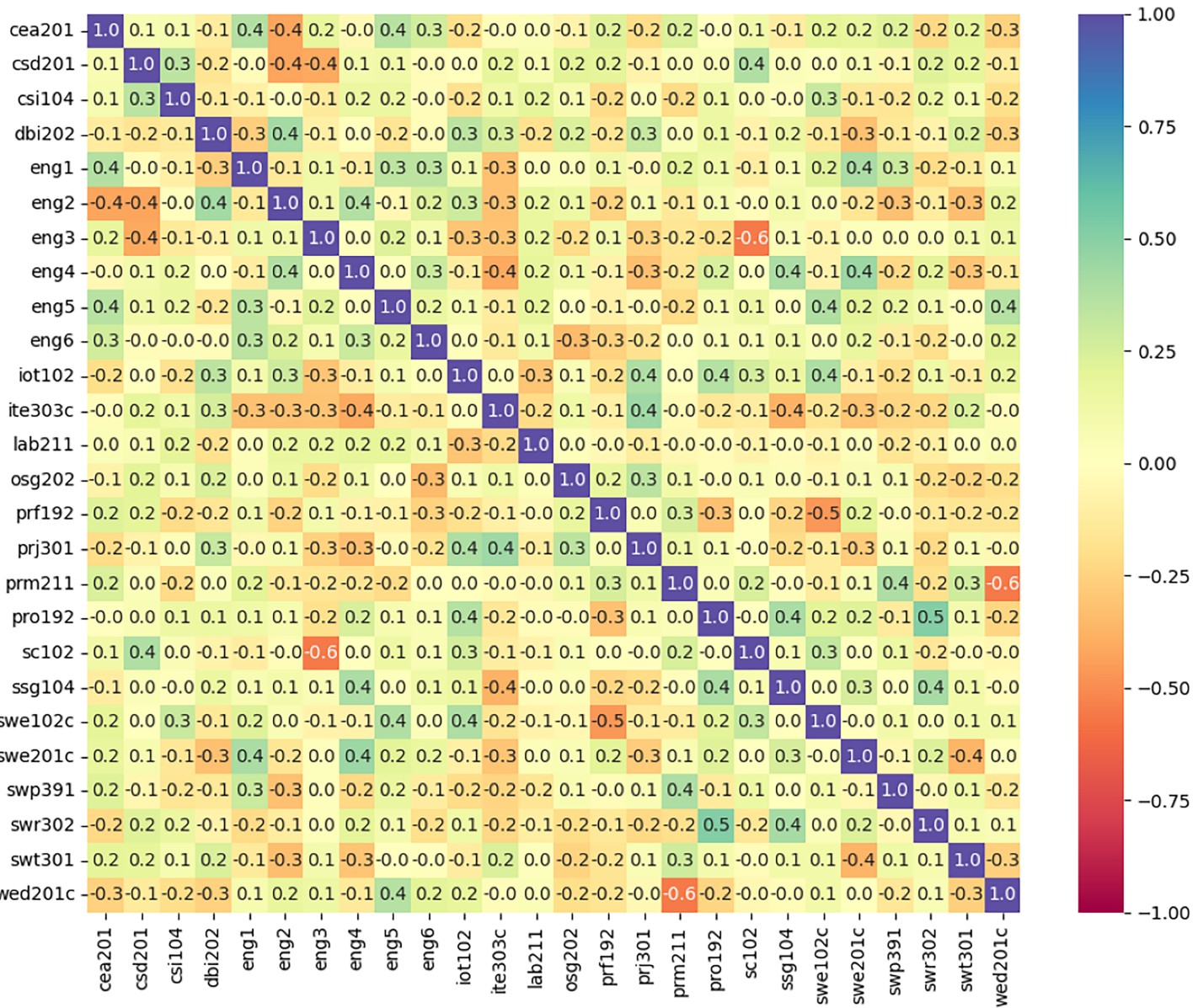

**Figure 3** The correlation between the features used in training model.               

The second stage also includes data normalized to ensures that all features are considered equally, and no single feature dominates the prediction process. Till second stage dataset consisted of 243 samples with 26 features extracted from different subjects of registered students who are enrolled for two academic years in university record. These features are considered as input variables which are used to predict output variable as dropout (presented by 0) and did not dropout (presented by 1) through proposed model.

In the third stage, data augmentation techniques are applied. The main objective of data augmenting is to capture the diverse characteristics of the dataset by generating new

samples. A variational autoEncoder (VAE) (*Kingma & Welling, 2014*) model is employed for augmenting the dataset. The VAE model generates synthetic data samples to increase the size of the dataset and thereby enhance its comprehensiveness (*Papadopoulos & Karalis, 2023*). Data augmentation artificially increases the size of dataset in which minor changes will be made in the original dataset to generate new data points. Although new data points are generated, it can be used to train the model and improve the quality of the data (*Li, Tai & Huang, 2019*).

After augmentation, the synthetic data is arranged and stored in a .csv formatted file for further processing. After augmentation, the number of records was increased to up to 100,000 data samples. From the overall data samples, 75% of the data was used for training and 25% for testing the performance of the DeepS3VM model. Training data is used for training the proposed model for predicting the dropout rate in the early stages and performance of the proposed hybrid model is validated using the testing data.

## RESULTS AND DISCUSSION

### Results

The performance of the proposed DeepS3VM model has undergone rigorous evaluation and comparison with existing models, using a comprehensive set of performance metrics. These metrics include accuracy, precision, recall, F1-score, and ROC curve (*Powers, 2011*). The existing models against which DeepS3VM has been benchmarked encompass a wide range of ML and DL models. In this research, the accuracy is measured using four different classification elements namely: true positives (TP), true negatives (TN), false positives (FP), false negatives (FN). These terms are used for constructing a confusion matrix.

The metrics used in the performance evaluation are computed as presented by:

$$\text{Accuracy} = \frac{TP + TN}{TP + TN + FP + FN} \tag{16}$$

$$\text{Recall} = \frac{TP}{TP + FN} \tag{17}$$

$$\text{F1-score} = \frac{2 * Precision * Recall}{Precision + Recall} \tag{18}$$

$$\text{Precision} = \frac{TP}{TP + FP}. \tag{19}$$

It must be ensured that the prediction model is said to be efficient if it exhibits a high prediction accuracy with a minimum error rate. The classification report of the proposed approach is tabulated in Table 5 and Fig. 4 illustrates the comparison analysis.

It has been observed from Table 5 that our proposed approach achieved the highest accuracy of 92.54% with the highest precision of 93.97%, recall of 96.75%, F1-score of 95.34% and an AUC score of 0.8679. Despite the initial limitations of the dataset (243 records), our approach utilized data augmentation to expand the dataset to 100,000 records and incorporated Apache Cassandra and Apache Spark for efficient dataset management and processing in predictive modeling. As a result, our model achieved robust performance across key metrics, demonstrating its effectiveness.

**Table 5 DeepS3VM model performance report.** The proposed model (DeepS3VM) with other models in term of methods, dataset, accuracy, precision, recall, F1-score, ROC score.

| Model trained | Dataset used | Authors | Accuracy | Precision | Recall | F1 score | ROC score |
|---|---|---|---|---|---|---|---|
| RF | 4,425 records and 35 feartures of students enrolled in various undergraduate degrees offered at a higher education institution | *El Aouifi, El Hajji & Es-Saady (2024)* | 77.6% | 92.08% | 78.33% | 84.65% | 0.7659 |
| DT | Collected 2,412 records of first-year students from a private university (UNI) in Taiwan | *Huynh-Cam, Chen & Lu (2024)* | 81.48% | 88.66% | 87.74% | 88.20% | 0.7294 |
| XGBoost | Dataset collected from Polytechnic Institute of Portalegre (IPP), Portalegre, Portugal | *Villar & de Andrade (2024)* | 89.13% | 92.44% | 93.89% | 93.16% | 0.8263 |
| ANN | Open University Learning Analytics (OULA) dataset | *Jiménez-Gutiérrez et al. (2024)* | 90.06% | 93.04% | 94.46% | 93.74% | 0.8405 |
| CNN | KDD Cup 2015 dataset | *Talebi, Torabi & Daneshpour (2024)* | 84.51% | 85.80% | 96.28% | 90.74% | 0.6843 |
| DeepS3VM | Collected from a private university in Vietnam, augmented dataset up to 100,000 records based on 243 samples with 26 features. | Proposed work | 92.54% | 93.97% | 96.75% | 95.34% | 0.8679 |

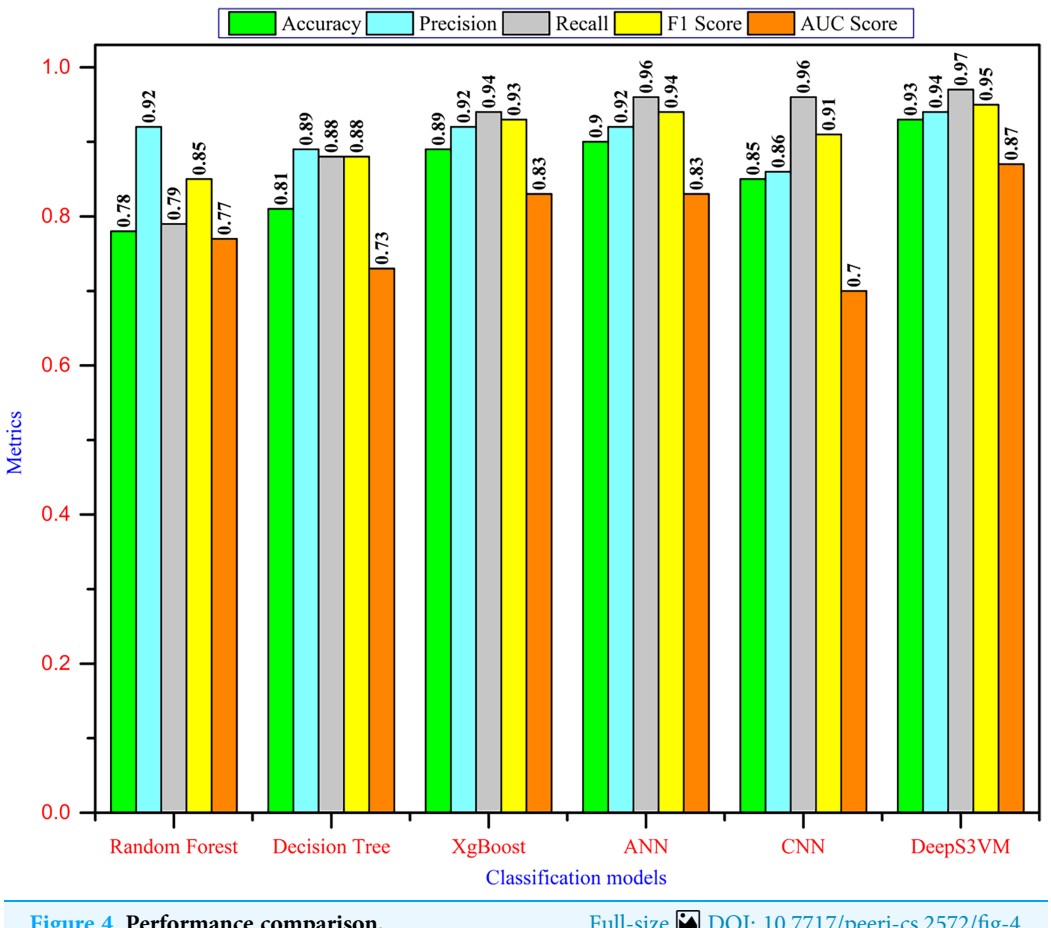

**Figure 4 Performance comparison.**

Figure 4 demonstrates that the DeepS3VM model outperforms all other models presented in the literature, highlighting its superior effectiveness and robustness. A glance at the chart reveals the Random Forest model performs well in terms of recall (0.92) and AUC (0.92), indicating that it is quite good at correctly identifying positive cases. However, its overall accuracy is comparatively lower (0.78), suggesting it might have more false positives or false negatives. In addition, the DT model shows a lower performance across most metrics. Its accuracy is the lowest (0.73), while recall and AUC scores are decent, but overall, it doesn't perform as well as the other models, which could indicate overfitting or limited generalization. XgBoost provides a balanced performance, with high precision, recall, and F1-score. Its accuracy (0.81) is better than DT and RF, indicating strong classification capabilities with relatively few false positives and negatives. The ANN model shows strong performance across all metrics, especially in terms of accuracy (0.92), F1-score (0.93), and AUC (0.93). This suggests that it's able to strike a good balance between precision and recall, and handle classification well, even on unseen data. CNN stands out as the top-performing model in terms of accuracy, precision, recall, and F1-score (all at 0.96), making it the most reliable classifier on this dataset. Its slightly lower AUC score (0.91) suggests that it might still struggle with some specific cases of class separation.

In contrast, the DeepS3VM model also performs very well, with recall (0.97) being the highest among all models. This shows that DeepS3VM is excellent at identifying true positives, which is critical in cases like student dropout prediction or other imbalance-sensitive problems. The model also excels in terms of accuracy (0.93), F1-score (0.95), and AUC (0.95), suggesting strong generalization and robustness.

To validate the effectiveness of the DeepS3VM model in terms of accurately classifying the requirements with very minimum false negatives, we have confusion matrices presented in different studies. Figure 5 compares the performance of DeepS3VM, Random Forest, XGBoost, Decision Tree, ANN, and CNN models in terms of true positives (correctly predicted positives) and false positives (incorrectly predicted positives). Regarding performance, DeepS3VM emerges as the strongest model with the highest true positives and the lowest false negatives, demonstrating its effectiveness in classifying both classes accurately. While, ANN and XGBoost also show competitive results, with a reasonable balance of true positives and false negatives. In addition, concerning the limitations in classification, RF and DT display higher false negatives, indicating potential challenges in correctly identifying positive cases, which may affect their effectiveness in prediction tasks. CNN, while having a high true positive rate, has a notably high false positive rate, which could compromise its precision. In conclusion, DeepS3VM is the most robust model in terms of both precision and recall, showing superior performance compared to other models in identifying true cases with minimal misclassification.

The performance of the DeepS3VM model is evaluated in terms of ROC and the output of the proposed model along with other models are shown in Fig. 6. These ROC curves represent the performance of the DeepS3VM model in comparison with other ML and DL models such as RF, DT, XGBoost, ANN and CNN. It classifies data into true positive rate (TPR) and false positive rate (FPR) classes. The ROC curve evaluates the trade-off between

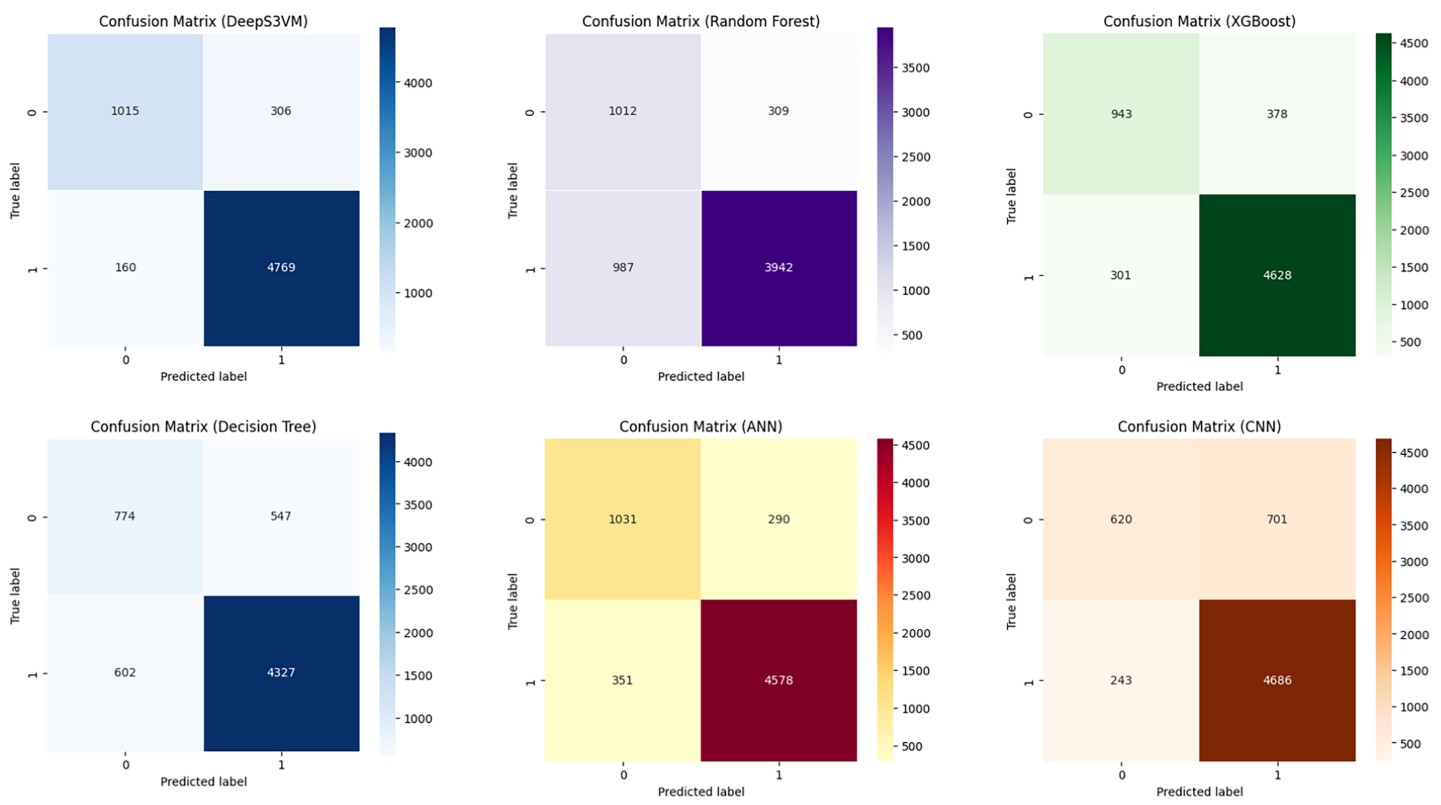

**Figure 5** The confusion matrix of models in research.

**Figure 6** ROC curve for performance evaluation.

**Table 6 Performance evaluation of the PRS.** The list of subject suggesting for students and comparing the performance with other models.

| ID | Dropout prediction rate (%) | | | Has recommended | | | Recommended subjects based on DeepS3VM |
|---|---|---|---|---|---|---|---|
| | DeepS3VM | ANN | RF | DeepS3VM | ANN | RF | 'CSI104' 'IOT102' 'ITE303c' 'ENGL5' 'ENGL1' 'SWT301' 'SWR302' 'ENGL3' 'SSG104' 'SSC102' |
| 1 | 99 | 99 | 100 | Yes | Yes | Yes | 'CSI104' 'ITE303c' 'ENGL5' 'IOT102' 'SWT301' 'ENGL1' 'SWR302' 'ENGL3' 'SSG104' 'SSC102' |
| 2 | 93 | 54 | 100 | Yes | Yes | Yes | 'CSI104' 'ITE303c' 'ENGL5' 'IOT102' 'ENGL1' 'SWT301' 'SWR302' 'ENGL3' 'SSC102' 'SSG104' |
| 8 | 94 | 98 | 100 | Yes | Yes | Yes | 'ITE303c' 'ENGL5' 'ENGL1' 'SWR302' 'SWT301' 'CSI104' 'IOT102' 'SSG104' 'ENGL3' 'SSC102' |
| 11 | 58 | 56 | 0 | Yes | Yes | No | 'ITE303c' 'ENGL5' 'ENGL1' 'SWR302' 'SWT301' 'CSI104' 'IOT102' 'SSG104' 'ENGL3' 'SSC102' |
| 49 | 99 | 0 | 0 | Yes | No | No | 'CSI104' 'ITE303c' 'ENGL5' 'IOT102' 'SWT301' 'ENGL1' 'SWR302' 'ENGL3' 'SSG104' 'SSC102' |

a model's sensitivity (recall) and specificity (precision) across multiple classification thresholds. Results show that the curve is closer to the top-left corner of the plot during the initial stages which represents higher sensitivity, high accuracy with lower FPR. The proposed approach achieves a ROC of 0.8679 which indicates that the hybrid DeepS3VM model has a superior ability to distinguish between positive and negative cases compared to all other models. It is higher than the ANN model (ROC of 0.84), XGBoost model (ROC of 0.826), RF model (ROC of 0.766), DT model (ROC of 0.729), and CNN model (ROC of 0.684). In summary, DeepS3VM is the most robust model in terms of ROC performance, with high true positive rates and low false positive rates, making it highly suitable for applications requiring accurate classification.

The results validate the effectiveness of the proposed approach. Since the combination of RNN and S3VM enhances performance by allowing RNN to focus on representation learning (extracting sequential features), while S3VM performs semi-supervised classification. This results in better generalization by leveraging both labeled and unlabeled data, improving the accuracy through refined decision boundaries. The semi-supervised nature of S3VM boosts precision by reducing incorrect classifications, while RNN's ability to capture temporal relationships improves recall, especially for instances that simpler models may miss. This synergy leads to better F1-scores, as it balances precision and recall, which is critical in handling imbalanced datasets (*e.g.*, dropout prediction). Furthermore, ROC-AUC scores benefit from RNN's ability to extract meaningful features, enabling S3VM to better distinguish between classes, even with sparse labeled data. This approach reduces false positives and false negatives, resulting in improved precision, recall, F1-score, and AUC, making it particularly effective for tasks with imbalanced data.

Moreover, the performance of the PRS is also evaluated by comparing it with other existing techniques and the results are tabulated in Table 6.

As inferred from the random results in Table 6, it can be said that the proposed PRS based on the DeepS3VM model is successful in recommending subjects in different scenarios compared to other models such as ANN and RF. For instance, for students 1, 2, 8, and 49, the DeepS3VM model recommends a similar set of subjects including 'CSI104', 'ITE303c', 'ENGL5', 'IOT102', 'ENGL1', 'SWT301', 'SWR302', 'ENGL3', 'SSG104', and 'SSC102'. This indicates that the DeepS3VM model is robust in its recommendations despite variations in dropout possibilities and other model's outcomes. The percentage of dropout possibility is different for other students. While some students exhibit high dropout possibilities according to DeepS3VM (*e.g.*, student 49 with 99% dropout prediction), the RF model sometimes predicts lower dropout possibilities or even zero. Such discrepancies suggest variations in the predictive capabilities of different models. This drawback is addressed by the DeepS3VM model since it provides consistent and accurate personalized recommendation for different student profiles.

## Discussion

This study presents significant advancements in the field of educational data analytics, particularly in predicting and mitigating student dropout. The DeepS3VM model demonstrates exceptional efficacy in predicting student dropout, achieving a recall of 0.97, thereby proving highly effective in identifying at-risk students, even within class-imbalance contexts. The model's strong performance metrics including accuracy (0.93), F1-score (0.95), and AUC (0.95), underscore its robustness and generalizability, positioning it as a valuable tool for early intervention strategies. Performance metrics of DeepS3VM surpass those documented in previous studies reviewed in the literature, highlighting the enhanced effectiveness of our approach. These results not only validate the DeepS3VM model as a powerful tool for early intervention strategies but also highlight its potential for broader applications in educational and predictive analytics, where early identification is essential for effective intervention.

Secondly, this research goes beyond prediction by proposing actionable insights through a prioritized list of courses for at-risk students. This recommendation is grounded in dropout prediction outcomes and offers a targeted approach to mitigate dropout risks, helping students focus on courses that may significantly impact their academic success. By aligning intervention with specific course recommendations, this study adds a layer of strategic support to dropout prevention efforts, bridging the gap between predictive insights and practical guidance. This approach is particularly valuable in educational settings where personalized learning paths can enhance student engagement and retention.

Finally, the study introduces an innovative hybrid data processing framework that combines Apache Cassandra and Apache Spark, providing a scalable and efficient solution for managing and processing large-scale datasets. This framework represents a methodological advancement, particularly suited for handling the unique challenges of educational data, such as imbalanced datasets and the risk of overfitting. Apache Cassandra's high-throughput data management paired with Spark's in-memory

processing capability allows for optimized data handling and efficient model training. This integration not only supports the scalability of predictive models but also facilitates real-time data processing, making the framework highly applicable in environments that demand both high throughput and low latency. The ability to address overfitting and imbalance challenges within this framework further enhances the reliability and accuracy of the dropout prediction model, contributing a novel solution for educational data processing needs.

While the advanced DeepS3VM system provides more accurate predictions of student dropout and offers targeted recommendations for subjects that require attention, it does have certain limitations. These include the complexity of the models, high computational costs, significant data requirements, and potential challenges related to generalization.

## CONCLUSION

Prediction of student drop out risk in the early stages of a course can be beneficial to both students and educational instructors. Deploying an accurate prediction model can help educational institutions to identify the students who are at the risk of dropout and assist them to address the problems and improve their learning behavior. In this context, this article designed a unique and efficient approach for predicting the risk of dropout based on the learning attributes. It was observed from the experimental analysis that the deployment of big data tools ominously enhanced the effectiveness of the proposed predictive model. A DeepS3VM model is designed by integrating SVM with RNN to maximize efficiency. Experimental outcomes reveal that the DeepS3VM model exhibited superior performance in terms of predicting the drop out percentage of at-risk students in the early stage of enrolment. The DeepS3VM model achieves an accuracy of 92.54% compared to other ML and DL models. The PRS model applied the results of prediction to recommends a learning path for students to avoid dropout. It was observed from the experimental analysis that the deployment of big data tools ominously enhanced the effectiveness of the proposed predictive model. In this way, the DeepS3VM model helps educational institutions and instructors in formulating an effective learning model, which in turn helps the students in making optimal decisions concerning their learning process.

For future model improvements, this study advocates for the integration of emerging data streams, such as social media activity and sensor-based information. In educational settings, sensor-based data can be obtained from devices embedded within learning environments, capturing a broad spectrum of data types, including students' emotional states and interaction patterns. These data streams offer valuable insights into various factors that influence the learning process, such as environmental conditions (*e.g.*, temperature and lighting) and instructional variables (*e.g.*, teacher' vocal characteristics). Moreover, the instructional approach and the level of teacher engagement should be taken into account when examining the issue of student dropout. This approach has the potential to deepen our understanding of how contextual elements impact student engagement and performance.

### Funding
The authors received no funding for this work.

### Competing Interests
The authors declare that they have no competing interests.

### Author Contributions
- Huong Nguyen Thi Cam conceived and designed the experiments, performed the experiments, analyzed the data, performed the computation work, prepared figures and/or tables, authored or reviewed drafts of the article, analysis tools, and approved the final draft.
- Aliza Sarlan conceived and designed the experiments, performed the experiments, authored or reviewed drafts of the article, contributed methodology, and approved the final draft.
- Noreen Izza Arshad conceived and designed the experiments, performed the experiments, authored or reviewed drafts of the article, contributed materials, and approved the final draft.

### Data Availability
   The raw data and code are available in the Supplemental Files.

### Supplemental Information
Supplemental information for this article can be found online at http://dx.doi.org/10.7717/peerj-cs.2572#supplemental-information.

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
