# Peer review of "A hybrid model integrating recurrent neural networks and the semi-supervised support vector machine for identification of early student dropout risk"

_PeerJ Computer Science, doi:10.7717/peerj-cs.2572_

## Round 0.1 · original submission · Major Revisions

Dear authors,

Thank you for submitting your article. Reviewers have now commented on your article and suggest major revisions. We do encourage you to address the concerns and criticisms of the reviewers and resubmit your article once you have updated it accordingly. When submitting the revised version of your article, it will be better to address the following:

1. In general, the literature review is not sufficient. More recent literature should be explored in depth. It is more of the type “researcher X did Y” rather than an authoritative synthesis assessing the current state-of-the-art. Advantages and disadvantages of the related works should be evaluated.
2. Pros and cons of the methods should be clarified. What are the limitation(s) methodology(ies) adopted in this work? Please indicate practical advantages, and discuss research limitations.
3. Many of the equations are part of the related sentences. Attention is needed for correct sentence formation.
4. Equations should be used with correct equation number. Please do not use “as follows”, “given as”, etc. Explanation of the equations should also be checked. All variables should be written in italic as in the equations. Their definitions and boundaries should be defined. Necessary references should be provided.
5. All of the values for the parameters of all algorithms selected for comparison should be given.
6. Discuss the limitations of your study.

Best wishes,

Reviewer 1 ·

Basic reporting

Authors use clear, unambiguous, professional English language.
The introduction and the background to the display of the context are described clearly enough, although an overview of the related articles of the last year (2022 - 2024) is missing. The introduction lacks a definition of the research/paper aim. Therefore, it is not clear what authors achieving.
Literature referenced & relevant; nevertheless, the review of the related articles of the last year (2022 - 2024) is missing. Moreover, the existing description of the related work is limited. It would be useful to extend Table 1 with columns such as attributes used for the prediction, dataset used for the prediction, etc. All those attributes could improve authors research and give more insights.
Structure of the article is clear.
Figures are relevant, well labelled & described. Some figures are fuzzy. Nevertheless, I suggest using appropriate notation for presenting the authors model (Figure 1). At the moment, the proposed model is missing clarity what is presenting: process or data. Figure 3 is duplicating already known data preprocessing steps.

Experimental design

In the methodology section, authors have presented a number of equations. However it is not clear which equations are new, and which are developed by authors. How equations are combined into one continuous formalization of the proposed model?
Some comments regarding the model description are provided in 1. Basic reporting. Please, see them.

Validity of the findings

Authors comparing their findings with other authors in Table 4. Nevertheless, can these different experiments be compared? Are they comparable?

At the beginning of the article, the authors talk about student dropout rates and reducing dropout rates. However, when conducting research, the authors forget what they are doing. This is all the research boils down to comparing different algorithms. I would like to get and see the full picture of how the authors adapt and use the proposed algorithm to reduce student dropout. Therefore, in its current state, the article is incomplete and the final results of how the authors' study helps to reduce student dropout rates are not presented. The whole case study is helpful here.

Regarding the conclusions, they are not based on the obtained results. Currently, only algorithms comparison is performed.

Additional comments

At the beginning of the article, the authors talk about student dropout rates and reducing dropout rates. However, when conducting research, the authors forget what they are doing. This is all the research boils down to comparing different algorithms. I would like to get and see the full picture of how the authors adapt and use the proposed algorithm to reduce student dropout. Therefore, in its current state, the article is incomplete and the final results of how the authors' study helps to reduce student dropout rates are not presented.

·

Basic reporting

Several sections need to be clarified and need to be rewritten. The paper needs to be proofread.
Sufficient background and context were provided. I think relevant literature was appropriately referenced.
The paper is well-structured overall. However, the figures and tables need to be reviewed. I did not find the raw data.

See the Additional Comments section for more details.

Experimental design

The research objectives are within the journal's Aims and Scope. The research question is well-defined, relevant, and meaningful. However, the methods described need to be defined more clearly, as in their current status, it is not possible to replicate the work proposed.

Validity of the findings

Since the methodology is unclear, it is very hard to assess the results produced. See the Additional Comments section for more details.

Additional comments

In their paper, the authors propose a hybrid model (DeepS3VM) to identify early student dropouts from a private university in Vietnam. As mentioned in the paper, student dropouts from college are critical, and higher education institutions worldwide are trying to minimize their effects by predicting students at risk and suggesting remedial actions to enhance their performance and help them graduate.
As far as I understand, the authors proposed a hybrid model called DeepS3VM, combining the RNN model with S3VM. However, the paper’s title would insinuate that the hybrid model is proposed based on the DeepS3VM technique. Hence, I suggest to review the title.
In line 226, the authors wrote, “This information is collected as a dataset of at-risk students that are further used to train the DeepS3VM model.” The authors claim that they collected a dataset related to “at-risk students.” Usually, the data (collected from the university database) contains students performing well and others identified as at-risk.
Table 2 needs more elaboration. It needs to clarify the difference between grades and assessment scores. How is the status of learning progress described?
In line 259, the authors wrote, “Parameters used to train RNN include historical academic records, such as grades, attendance, participation in extracurricular activities, and demographic information.” However, the ‘participation in extracurricular activities’ feature is not mentioned in Table 2.
The columns in Table 3 are not related. I suggest re-designing this table to remove ambiguity. In addition, In this Table, the authors enumerated the features selected, which I assume are the labeled data. What kind of unlabeled data is used in the proposed model?
From Figure 2, we can understand that the output of the RNN model is the input to the S3VM model. However, the process described in the implementation section insinuates that the data is divided at the beginning into two sets: labeled (for the RNN model )and unlabeled data (for the S3VM model). The whole idea needs to be clarified, and I recommend rephrasing it.
In line 533, the authors stated, “the performance of the proposed model DeepS3VM with other models proposed in the literature to prove that our proposed model out-performs all other models proposed in the literature.” It is unclear if the “other models” were applied to the same dataset replicating the proposed model, and the values presented in Table 4 and Fig. 5 are the output of this application. Again, in Table 5, the authors compare the proposed DeepS3VM, ANN, and RF results.
The authors did not mention any limitations to their work or future model enhancement.
Finally, I recommend to proofread the paper to enhance its readability.

---

## Round 0.2 · Minor Revisions

Dear authors,

Thank you for submitting your revised manuscript. Feedback from the reviewers is now available. It is still not recommended that your article be published in its current format. However, we strongly recommend that you address the minor issues raised by the reviewers and resubmit your paper after making the necessary changes.

Best wishes,

Reviewer 1 ·

Basic reporting

The paper is improved according to the comments.
Clear English used throughout.
Literature references are sufficient. Nevertheless, in Table 1 they can be described better, i.e., in more details, like drawbacks can be mentioned
Article structure, tables, raw data shared are sufficient. The quality of figures differs and have to be improved . Some of them are enlarged to much, others are to small.
Author should highline their contribution in the Introduction section. Now, It is not clear enough.

Experimental design

Original primary research within Aims and Scope of the journal.
Research question is not defined. How does the research fill an identified knowledge gap?
Rigorous investigation performed to a high technical & ethical standard.
Methods described with sufficient detail & information to replicate.

Validity of the findings

All underlying data have been provided; they are robust, statistically sound, & controlled.
Conclusions are well stated, linked to original research question & limited to supporting results.

Additional comments

Discussion should be in the separate section. Authors should discuss in details how the obtained results help in solving initially identified knowledge gaps.

Reviewer 3 ·

Basic reporting

In the study, a hybrid model called DeepS3VM, which is created by integrating recurrent neural networks and self-supervised support vector machine method, is presented, which identifies the school dropout risk of students at early stages and gives necessary recommendations to students with this risk.

The study is written in clear and understandable English.
It is well structured theoretically. The results are presented in detail with figures and tables.
References are sufficient and up to date.

Experimental design

More detailed information should be provided about the parameters in the raw dataset. This will allow a clearer understanding of which parameters are used to identify students at risk of dropout.

It has been stated that after the datasets are collected, duplicate and missing records are removed. Is removing missing records a correct solution? Doesn’t this cause loss of information? Why was removing this data considered a correct action and carried out? Can’t another solution be offered?

The proposed model has achieved higher success than all models in terms of accuracy, precision, recall, F1 score and ROC score. However, the reason for this success has not been explained. The results have not been analyzed in detail. The factors that cause the performance increase should be stated. The proposed method should be compared with other methods in more detail. In addition, the results given with figures should be interpreted better.

In the study, the proposed approach has been compared with many methods. It should be emphasized whether the results are obtained from the same data.

Validity of the findings

Comments mentioned in the experimental design already

Additional comments

No comment

---

## Round 0.3 · accepted · Accept

Dear Authors,

Thank you for addresding all of the reviewers' comments. The paper is now ready for publication.

Best wishes,

Reviewer 1 ·

Basic reporting

The authors have improved the paper sufficiently. No comments. Thank you.
Clear and unambiguous English is used throughout. Only some warnings are found, such as missing break after the dot in the conclusions.
Literature references are sufficient.
Article structure, figures, tables are sufficient. Raw data shared.

Experimental design

Research is within Aims and Scope of the journal.
Research question well defined, relevant & meaningful.
Investigation is performed to a technical & ethical standard.
Methods are described with sufficient detail & information to replicate.

Validity of the findings

Conclusions should be better linked to the research questions.

Additional comments

None.

·

Basic reporting

The paper is now written in a clear and unambiguous English. It is well structured.
Figures and Tables are sufficient. References are also adequate.

Experimental design

The authors addressed all the comments raised by the reviewers. The research questions are now well defined, The methods are described clearly.

Validity of the findings

Discussion and Conclusions are well stated.

Additional comments

No comments

Reviewer 3 ·

Basic reporting

No comment

Experimental design

Necessary changes have bben made in line with the comments in the first revision. The results and figures have been analyzed in more detail and better interpreted. The situations expressed as uncertainties have been explained.

Validity of the findings

No comment

Additional comments

No comment